# PreS1 Mutations Alter the Large HBsAg Antigenicity of a Hepatitis B Virus Strain Isolated in Bangladesh

**DOI:** 10.3390/ijms21020546

**Published:** 2020-01-15

**Authors:** Md. Golzar Hossain, Md. Muket Mahmud, K. H. M. Nazmul Hussain Nazir, Keiji Ueda

**Affiliations:** 1Division of Virology, Department of Microbiology and Immunology, Graduate School of Medicine, Osaka University, Osaka 565-0871, Japan; 2Department of Microbiology and Hygiene, Bangladesh Agricultural University, Mymensingh 2202, Bangladesh; muketvet@gmail.com (M.M.M.); nazir@bau.edu.bd (K.H.M.N.H.N.)

**Keywords:** hepatitis B virus (HBV), Bangladesh, full genome sequence, preS1 mutations, antigenicity

## Abstract

Mutations in the hepatitis B virus (HBV) genome can potentially lead to vaccination failure, diagnostic escape, and disease progression. However, there are no reports on viral gene expression and large hepatitis B surface antigen (HBsAg) antigenicity alterations due to mutations in HBV isolated from a Bangladeshi population. Here, we sequenced the full genome of the HBV isolated from a clinically infected patient in Bangladesh. The open reading frames (ORFs) (P, S, C, and X) of the isolated HBV strain were successfully amplified and cloned into a mammalian expression vector. The HBV isolate was identified as genotype C (sub-genotype C2), serotype adr, and evolutionarily related to strains isolated in Indonesia, Malaysia, and China. Clinically significant mutations, such as preS1 C2964A, reverse transcriptase domain I91L, and small HBsAg N3S, were identified. The viral P, S, C, and X genes were expressed in HEK-293T and HepG2 cells by transient transfection with a native subcellular distribution pattern analyzed by immunofluorescence assay. Western blotting of large HBsAg using preS1 antibody showed no staining, and preS1 ELISA showed a significant reduction in reactivity due to amino acid mutations. This mutated preS1 sequence has been identified in several Asian countries. To our knowledge, this is the first report investigating changes in large HBsAg antigenicity due to preS1 mutations.

## 1. Introduction

Hepatitis B virus (HBV) is one of the smallest known animal viruses; it is known to cause acute and chronic hepatitis in humans, followed by liver cirrhosis and hepatocellular carcinoma (HCC). The mature infectious virion (Dane particle, approximately 42 nm in diameter) consists of an icosahedral nucleocapsid surrounded by three kinds of envelope proteins [1]. HBV is distributed globally with abundant mutations in its genome due to the lack of proofreading capacity in the viral polymerase protein; this genomic variability has led to what is known as quasispecies. These genomic variations show a strong association with the clinical characteristics of HBV-infected patients and geographical distribution [2]. HBV has been categorized into nine genotypes (A–I) depending on nucleotide variations in at least 8% of the total genome in different strains [3]. However, the partially double-stranded 3.2 kb viral genome encodes four overlapping open reading frames (ORFs): P, S, C, and X. Three kinds of envelope/surface proteins are encoded by a single ORF, S, because it contains three in-frame start codons in the preS1, preS2, and S regions. The large (preS1+preS2+S), medium (preS2+S), and small (S only) surface proteins share a common C terminus region [4].

The core protein plays important roles during viral replication and interacts with several host cellular factors [5,6,7]. HBx, encoded by ORF X, is a diverse and multifunctional nonstructural protein associated with many cellular signaling pathways; it plays a vital role in the development of HBV-induced HCC [8,9]. HBV Pol is a multifunctional protein with reverse transcriptase activity that lacks proofreading capacity during its translation. Therefore, many drug-resistant mutations in the reverse transcriptase (RT) domain of Pol have been reported, indicating the need for novel anti-HBV drugs [10]. On the other hand, the coding regions of all the surface proteins of HBV, including their promoters (preS1 and S), fully overlap with the terminal protein (TP) domain and RT regions of Pol [11,12]. Hence, mutations in these regions can also be found in the surface proteins.

Although the small hepatitis B surface antigen (HBsAg) plays an important role in the diagnosis of acute infection using commercial ELISA kits, mutations are known to lead to diagnostic failures [12]. HBV infection may also be diagnosed through the detection of the medium and large HBsAg proteins using commercial ELISA kits [13,14,15]. Anti-preS1 antibodies may be the first serological marker of the immune response against HBV infection, as it occurs in the early phase of infection because of its location on the surface of the HBV virion [16]. However, mutations may also occur in the medium and large surface proteins, and mutations in the preS1 region are reported to be associated with HCC [17,18,19]. HBV enters the host cell through the binding of preS1 with the sodium taurocholate co-transporting polypeptide (NTCP) receptor [20]. The preS1 region contains several immunogenic domains and T- and B-cell epitopes [21]. Recently, the development of an HBV vaccine targeting preS1 has been attempted [22]. However, it has not been confirmed whether changes in antigenic properties due to these mutations affect the efficacy of the vaccine. The development/modification of novel/available diagnostic kits/HBsAg ELISA kits are required and require routine mutational analyses of currently circulating HBV genomes in different populations.

Bangladesh is a densely populated country with a high prevalence of HBV infection and HBV-induced HCC [23]. HBV genotype C is the most prevalent subtype in the Indian subcontinent, including in Bangladesh [24,25,26], and is thought to be more harmful and more likely to lead to HCC [27,28,29]. Drug resistance and HBsAg diagnostic escape mutations have been also reported in Bangladesh based on HBV genome sequence analysis only [24,25,26,30,31]. In this study, we sequenced the complete genome of HBV isolated from an acute case in Bangladesh and compared its sequence with those of previously published sequences. Next, we cloned the major HBV viral protein-coding sequences to an expression vector and analyzed the expression and distribution of these viral proteins in HEK-293T and HepG2 cells by immunofluorescence analysis. Finally, we performed ELISA and western blot analysis of the small and large HBsAg proteins to determine the effect of mutations in the preS1 coding region on its antigenicity.

## 2. Results

### 2.1. Confirmation of HBV Infection/Replication

The serum sample was collected from a patient with a history of jaundice who was confirmed to be HBsAg-positive (OD_450/630_: 3.66; cut-off value: 0.105) and showed a high level of HBeAg (OD_450/630_: 2.44; cut-off value: 0.105). qPCR data showed active viremia with a titer of 1.16 × 10^9^ copies viral genome/mL serum. These data showed that active viral replication occurred in the liver of the infected patient; this also verified that the patient had no history of treatment with anti-HBV drugs.

### 2.2. Successful Amplification and Cloning of Four Full Viral Genes

We successfully amplified all four viral genes: P, S, C, and X, which have a length of 2532 bp, 1203 bp, 552 bp, and 465 bp, respectively. Then, we cloned the ORFs of each gene into the mammalian expression plasmid pCMV-Myc. The correct fragment sizes containing each clone in pCMV-Myc plasmid were screened after digestion with restriction enzymes (EcoRI and SalI). At least two correct clones for each ORF were selected for further confirmation by sequencing using CMV promoter primers. Sequencing data confirmed the ligation of the correct in-frame specific ORFs with Myc tag; both clones showed identical nucleotide sequences with the sequences from direct PCR products as well. We further confirmed the fragment size of each ORF through a comparison with our previously cloned HBV ORFs in the same expression plasmid. These results suggest that the isolated HBV genes could be easily amplified by PCR and showed no truncated deletions after cloning into the expression plasmids.

### 2.3. Characterization, Evolutionary Origin, and Mutational Analysis

The total genome length of the isolated HBV strain in this study was 3215 bp. Identical nucleotide sequences were obtained from direct PCR products of viral genomic DNA and HBV ORFs cloned into pCMV-Myc plasmids. The HBV genome sequence was annotated using a previously published Bangladeshi HBV full genome (MF925359.1) and a reference HBV strain used in our laboratory (X01587). The isolated HBV in this study (BD2) was confirmed to be genotype C and sub-genotype C2, as determined by Geno2pheno:hbv and serotype (subtype) adr. Phylogenetic trees can be used to reveal the evolutionary origins of microorganisms and are commonly used to analyze HBV evolution patterns [24,25,32]. Our phylogenetic tree, constructed with different HBV genotypes obtained from GenBank, showed that BD2 was clustered with Bangladeshi HBV isolates, which are closely related to HBV genomes originating from Indonesia, Malaysia, and China (Figure 1). However, the maximum sequence identity of 99.47% was obtained in an HBV isolate originating in Thailand (KM999990.1). Mutational analysis at the nucleotide level showed only two mutations, C2964A and C3062T, in the preS1 region of BD2 HBV; no mutations were found in the preS2 region, consistent with previous findings (Figure 2) [25]. No nucleotide mutations were identified in the basal core promoter (BCP) or PC/core promoter. Only one putative NA-resistant mutation, I91L, was found, and no primary, secondary, pre-treatment, or drug-resistant mutations were observed in the RT domain of Pol. However, some mutations (H9Y, N13H, I91L, I269L, and V278I) were found in the RT domain that are not associated with drug resistance, according to Geno2pheno:hbv. Small HBsAg showed the following mutations: N3S, V18G, E44G, M47T, S53L, V159A, A177V, S210N, and I213L; none of these are associated with HBsAg escape, though the N3S mutant is associated with an increased risk of HCC [25].

### 2.4. Expression Analysis of Different HBV Proteins

Mutations in HBV proteins may affect their expression and subcellular distribution; currently, there are no reports of expression analysis of viral proteins in Bangladeshi HBV isolates. Therefore, we analyzed the expression of viral proteins in vitro by transient transfection in HEK-293T and HepG2 cells. The expression and subcellular distribution of polymerase, large HBsAg, core, and HBx were confirmed by immunofluorescence assay (Figure 3). The subcellular distribution of the viral proteins showed similar staining patterns in both HEK-293T and HepG2 cells to that of our reference strain (adr4, X01587), despite the presence of several mutations in BD2 (MK628732). These results suggest that the major HBV proteins isolated from Bangladesh were successfully expressed in a cell culture system with native subcellular distribution.

### 2.5. Antigenicity of Large HBsAg Is Altered due to Mutations in preS1 Region

Amino acid mutations may change the conformation of HBsAg [12]. Therefore, we compared the antigenicity of BD2 with that of the reference HBsAg adr4 from transfected HEK-293T and HepG2 cell lysates using HBsAg ELISA. The reactivity of BD2 is slightly lower. Nine mutations were found in the preS1 domain of large HBsAg from BD2 HBV isolates compared to that from the reference strain adr4 (Figure 4A). Therefore, we tested the antigenicity of large HBsAg from the transfected HEK-293T and HepG2 cells using preS1 ELISA. Interestingly, BD2 large HBsAg showed significantly reduced binding capacity with preS1 ELISA (Figure 4C,H). To confirm this, we performed western blotting from both transfected cell lines (HEK-293T and HepG2) using two different antibodies: one targeting the Myc tag, and the other targeting preS1 with a commercially available antibody. The Myc tag antibody showed comparable levels of large HBsAg expression between both groups, whereas the anti-preS1 antibody could not detect large HBsAg in the BD2 samples (Figure 4D,E,G). This suggests that amino acid mutations in the preS1 region affected the antigenicity of large HBsAg.

### 2.6. Silent Large HBsAg Containing Mutated HBV Is Circulating in Asiatic Countries

We found several mutations in the preS1 region that may be responsible for antigenic alterations in large HBsAg. Thus, we investigated whether these mutations in the HBV genome were present in sequences deposited in the NCBI database from other countries. We searched in BLAST using 119 amino acids/357 nt of the preS1 region of BD2 genome and found a total of 103 amino acid sequences and 60 nucleotide sequences showing 100% sequence identity. These preS1 regions mutations were found in HBV genomes isolated from Asian countries, including Thailand, Myanmar, Cambodia, Laos, Malaysia, India, Bangladesh, Indonesia, and Japan (Figure 5A,B). These results indicate that HBV strains containing this mutated large HBsAg are circulating among these countries.

## 3. Discussion

HBV is a serious public health problem worldwide, including in Bangladesh. Bangladesh is a densely populated country with a high prevalence of HBV and a predominance of subtype C/C2 [24,25,26]. HBV mutations might affect the success rates of diagnostic/vaccination protocols, leading to the development of drug-resistant strains [33,34,35,36,37,38,39]. The expression, distribution, and secretion of HBV proteins could be affected by amino acid mutations that are also correlated with HCC [40,41,42,43]. Here, we identified an HBV strain from an acute clinically infected patient and performed full genome sequencing, characterization, mutational analysis, cloning, and expression analysis of the major viral proteins. Several mutations in the preS1 region were found that alter the antigenicity of large HBsAg against antibodies; furthermore, this HBV strain containing silent antigenic large HBsAg mutations is circulating in Asian countries.

The detection of HBsAg is the primary marker of acute HBV infection, and active viral replication is indicated based on the detection of HBeAg and serum DNA levels [44,45]. Genetic variations in HBV, as well as recombination between different genotypes determine its severity, as well as the progression to HCC [46]. The evolutionary analysis of the whole genome sequence of Bangladeshi HBV isolates showed a close relationship with those from neighboring countries such as India, Myanmar, Nepal, and Thailand, as well as high recombination rates [25]. PreS1 region mutations might be related to the progression of liver diseases, and these mutations have been reported in multiple HBV genomes isolated in Bangladesh [25]. Polymerase mutations potentially lead to drug resistance, which is a major cause of chronic hepatitis or HCC due to the ineffectiveness of anti-HBV drugs [47]. Some RT mutations, such as rtI91L, have been positively associated with HCC; however, this has not been experimentally confirmed in vitro and is considered a putative nucleotide analogues-resistant mutation. These mutations have been reported in Bangladesh as well [25,48].

The HBV genome encodes four major proteins, and each of them has a different function. Their expression patterns in hepatocyte-derived cells vary in infection compared to transfection contexts. However, no expression analysis of the viral proteins of a Bangladeshi HBV isolate has been performed so far. HBV Pol and core are known to localize to the cytoplasm in transfected cells, with Pol showing subcellular distribution near the mitochondria; similar results were found in this study [49,50]. HBV envelope proteins are expressed in the cytoplasm, post-translationally glycosylated into the endoplasmic reticulum (ER) and Golgi bodies, though fractions of these proteins may localize into mitochondria [51,52]. Mutations in HBsAg may lead to changes in its subcellular distribution and staining patterns [53]. HBx is a non-structural protein that plays significant roles in the development of HCC; it is expressed in the cytoplasm and/or nucleus [54,55]. Although the HBV proteins isolated from Bangladeshi patients displayed several mutations, expression analysis of our viral proteins by IFA in HEK-293T and HepG2 cells showed a high similarity to native HBV expression properties.

The preS1 region plays an important role in NTCP receptor-mediated HBV infection; mutations in this region have been associated with HCC [19]. Yin et al. identified HBV with a C2964A mutation in the preS1 region as a novel factor associated with HCC [56]; this mutation has also been reported in the present study. The preS1 region (residues 1–47) is necessary for binding with the NTCP receptor for successful infection [57,58]. Thus, a preS1 peptide-based vaccine could be an alternative strategy to prevent HBV infection [22,59]. Therefore, whether the antigenicity of large HBsAg due to mutations in the preS1 region would affect diagnostic accuracy using commercial large HBsAg ELISA kits was investigated here. In the current report, nine mutations were found in our isolate when compared to adr4 (X01587); these mutations led to a complete loss of reactivity with a commercial anti-preS1 antibody in western blot analysis. Moreover, the reactivity against PreS1 ELISA also showed significant reductions. PreS1 residues 21–59 are recognized as a major immunogenic domain; more precisely, preS1 residues 27 and 39 are involved in the formation of B- and T-cell epitopes. In addition, the D27G mutation has been identified in patients with active chronic hepatitis [21,60,61,62]. In this study, four mutations (D27G, H39N, H51Q, and E54A) were identified within residues 21–59. Furthermore, the NTCP-binding region of preS1 showed two mutations (D27G and H39N) that require further investigations into whether these mutations affect the HBV infection efficiency. These mutations, which are responsible for silent changes in the antigenicity of large HBsAg, have been identified in many Asian countries, including Bangladesh. However, the expression and secretion efficiency of envelope proteins and the identification of specific mutations responsible for preS1 silent antigenicity must be further investigated in future.

## 4. Materials and Methods

### 4.1. Sample Collection

The serum sample was collected from a 20-year-old HBsAg-positive male patient with a history of acute infection. Written consent has been taken from the patient mentioning that the sample will be used for research purposes and approved by Bangladesh Agricultural University research ethical committee [Approval ID: AWEEC/BAU/2019(47)]. Infection was confirmed by HBsAg and HBeAg ELISA in the laboratory.

### 4.2. DNA Extraction, PCR Amplification, and Cloning

Viral DNA was extracted from serum samples using the QIAamp DNA Mini Kit (Qiagen, Hilden, Germany) following the manufacturer’s protocol. Plasmid DNA from *Escherichia coli* DH5α was extracted using the PureYield Plasmid Maxiprep System and Wizard Plus Minipreps DNA Purification System (Promega, Madison, WI, USA) following the manufacturer’s guidelines. The full HBV genome (P, S, C, and X ORFs) was amplified by PCR using TaKaRa ExTaq polymerase (TaKaRa Bio, Shiga, Japan) and ORF-specific primer sets (Table 1). Forward and reverse primers were designed to include the cutting sites of the indicated restriction enzymes upstream of the start codons and downstream of the end codons for each ORF. PCR was performed at a final volume of 20 μL according to the following parameters: 96 °C for 5 min, followed by 35 cycles of 96 °C for 30 s, 54 °C for 30 s, and 72 °C for 2 min, with a final extension step at 72 °C for 7 min. The ORFs amplification was confirmed by 2% TBE agarose gel electrophoresis to justify the band size corresponded to the length of each specific gene. The PCR products were then purified using the MonoFas DNA Purification Kit (GL Sciences Inc., Tokyo, Japan) and digested with restriction enzymes (EcoRI and SalI). Then specific fragments of each gene (P, S, C, and X) were run on a 1% TAE agarose gel electrophoresis, purified, and ligated with pCMV-Myc (Clontech) following the standard protocols. The ligated plasmids were transformed into DH5α *E. coli*, and fragments containing the ligated pCMV-Myc plasmids were screened by digesting with restriction enzymes (EcoRI and SalI) after the extraction of plasmid DNA. Two successful clones for each ORF were selected for further confirmation by sequencing using CMV promoter primers.

### 4.3. Real-Time Quantitative PCR

The viral load in the serum was determined by quantitative PCR (qPCR) using Fast SYBR™ Green Master Mix (Applied Biosystems, Vilnius, Lithuania) with the QuantStudio 6 Flex qPCR machine (Applied Biosystems) following the manufacturer’s guidelines. qPCR was performed as previously described with a final concentration of 200 nM HBV-specific primers (Table 1) [63].

### 4.4. Sample Preparation for Sequencing

The direct products of PCR-amplified full ORFs (P, S, C, and X) and cloned plasmids (pCMV-Myc_HBV_ORFs) were subjected for sequencing using ORF-specific primers and CMV promoter primers respectively. Samples were prepared for sequencing by PCR at a final volume of 20 μL using BigDye^®^ Terminator v3.1 Cycle and BigDye^®^ Terminator v1.1 & v3.1 5× Sequencing Buffer (Applied Biosystems). PCR was performed according to the following parameters: 94 °C for 5 min, followed by 35 cycles of 94 °C for 30 s, 54 °C for 30 s, and 60 °C for 2 min, with a final extension step at 60 °C for 7 min. PCR products were precipitated with ethanol, dried, and dissolved in Hi-Di Formamide (Applied Biosystems). Then samples were boiled for 5 min and sequenced using the Applied Biosystems 3730 DNA Analyzer (California, CA, USA).

### 4.5. Sequence Annotation and Analysis and Phylogenetic and Mutational Analysis

First, the raw sequence data were edited, annotated, and analyzed using CLC Sequence Viewer (http://www.clcbio.com) and compared with two reference sequences (MF925359.1 and X01587). The genotype and sub-genotype were determined using Geno2pheno:hbv (https://hbv.geno2pheno.org/). The serotype was determined as previously described [64]. The BD2 HBV isolate was then submitted to GenBank (accession number: MK628732). ExPASy was used to obtain the amino acid sequence and for ORF in-frame check with Myc tag in pCMV-Myc vector (http://web.expasy.org). For the evolutionary analysis, a BLAST analysis was performed using the full BD2 genome to determine nucleotide identities with previously published HBV genomes. The BD2 HBV genome sequence was aligned with different genotypes and sub-genotypes for the construction of phylogenetic trees as previously described [25]. The phylogenetic tree was constructed using BD2 and reference sequences using MEGA software and CLC Sequence Viewer as previously described [24,25]. Mutational analysis of different viral proteins was performed using Geno2pheno:hbv (https://hbv.geno2pheno.org) and the HIV-Grade:HBV-Tool (https://www.hiv-grade.de/cms/grade/explanations/hbv-tool/). Nucleotide mutations at different regions, such as preS1, basal core promoter, and the RT region, were analyzed according a previously published report [25]. The preS1 region (nucleotides 1–357 and residues 1–119) was used a query sequence to search BLAST nucleotides and BLAST protein for sequences with 100% sequence identity.

### 4.6. Cell Lines and Transfection

HEK-293T cells and a human liver carcinoma cell line, HepG2, were used to express the major HBV proteins P, S, C, and X. Cells were originally obtained from ATCC and maintained and cultured in our laboratory according to our previous protocol [12]. The HEK-293T cells were seeded in 6-well collagen-coated plates (2 × 10^5^ cells/well) and incubated overnight. Transient transfection was performed using 1 μg plasmid using TransIT-LT1 reagent (Mirus Bio, Madison, WI, USA) according to the manufacturer’s guidelines. HepG2 cells were seeded similarly (5 × 10^5^ cells/well) and transfected using GenJet™ In Vitro DNA Transfection Reagent ver II (SignaGen Laboratories) according to the manufacturer’s guidelines. After 48 h, cells were collected for further experiments. Transfection for immunofluorescence staining was performed following our previous protocol [12].

### 4.7. Western Blot, Immunofluorescence Analysis, and ELISA

Western blotting and immunofluorescence analysis were performed on the transfected cells according to our previously described methods [12,63]. The transfected cells were lysed with 50 mM NaH_2_PO_4_ [pH: 8.0], 300 mM NaCl, 0.1% NP40, and complete mammalian protease inhibitor (Sigma P8849; 1:1000 dilution), and prepared lysates were used for ELISA and western blotting. Antibodies targeting the Myc tag (Anti-Myc tag antibody [9E10]; Millipore) and the preS1 (Anti HBV Pre-S1 antibody, mouse mono-1, Catalog No. BCL-AB-01; Beacle, Inc.) were used to detect large HBsA by western blot. ELISA for the detection of small HBsAg (HBsAg ELISA rapid II, Beacle Inc., Kyoto, Japan) from serum samples or lysates of transfected cells were performed according to the manufacturer’s guidelines. For ELISA of the large HBsAg (HBs Pre-S1 Quantitative ELISA Kit, Rapid; Beacle, Inc.) and E antigen (HBeAg ELISA Kit, Bioneovan, Beijing, China), the following modified protocol was used. First, 100 μL sample was loaded onto a well and incubated overnight at room temperature with gentle shaking. Then, 50 μL HRP-conjugated preS1 Ab or HBe Ab was added, and the samples were incubated for 1–2 h at room temperature with gentle shaking. The well was then washed five times and dried by shaking and with paper towels. The color reaction was developed by incubation with substrate solution A and B, and the reaction was stopped with stop solution. The optical density (OD_450_) was measured at 450 nm and quantified relative to that of the reference wavelength (OD_630_) using a Spectra Max 190 microplate spectrophotometer (Molecular Devices, San Jose, CA, USA).

### 4.8. Statistical Analysis

Statistical significance of antigenicity alteration was determined using *t*-test, and probability values of *p* ≤ 0.05 were considered statistically significant. Each experiment was performed at least three times, and the results were presented as the mean ± standard deviation (SD) of the mean.

## 5. Conclusions

An HBV strain isolated from an acute case in a Bangladeshi patient was characterized as genotype C (subtype C2) and serotype adr and was confirmed to be evolutionarily related to HBV strains from Indonesia, Malaysia, and China. preS1 C2964A, RT domain I91L, and small HBsAg N3S mutations were found, all of which have clinical importance. The four viral genes were successfully amplified by PCR and cloned in mammalian expression vectors. The major viral proteins all showed native expression properties. Mutations in the preS1 region altered the antigenicity of large HBsAg and are circulating in Asian countries. To the best of our knowledge, this is the first study reporting a clinical case of this mutated HBV strain showing large HBsAg antigenic alteration.

## Figures and Tables

**Figure 1 ijms-21-00546-f001:**
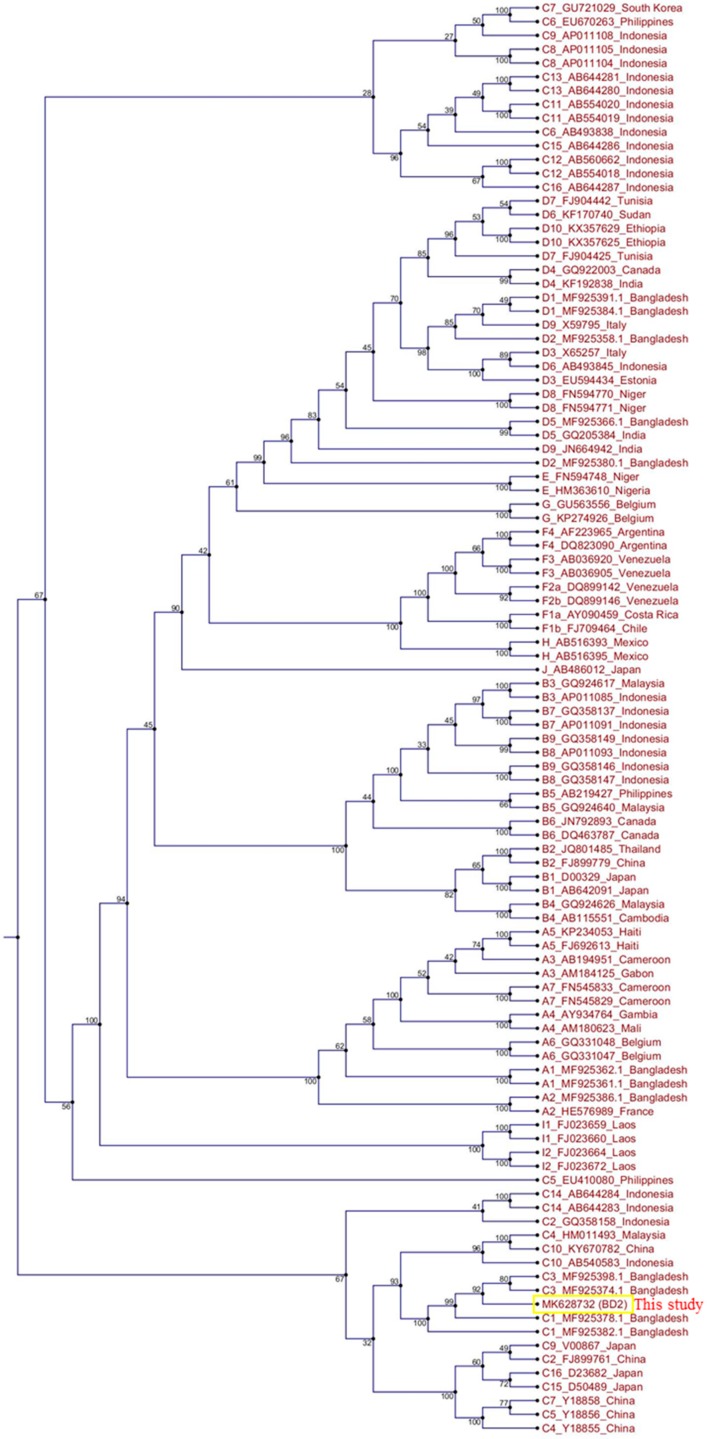
Phylogenetic tree analysis of hepatitis B virus (HBV) BD2 (MK628732). The full genome sequences of several HBV strains were extracted from GenBank, and a phylogenetic tree was constructed using CLC Sequence Viewer.

**Figure 2 ijms-21-00546-f002:**
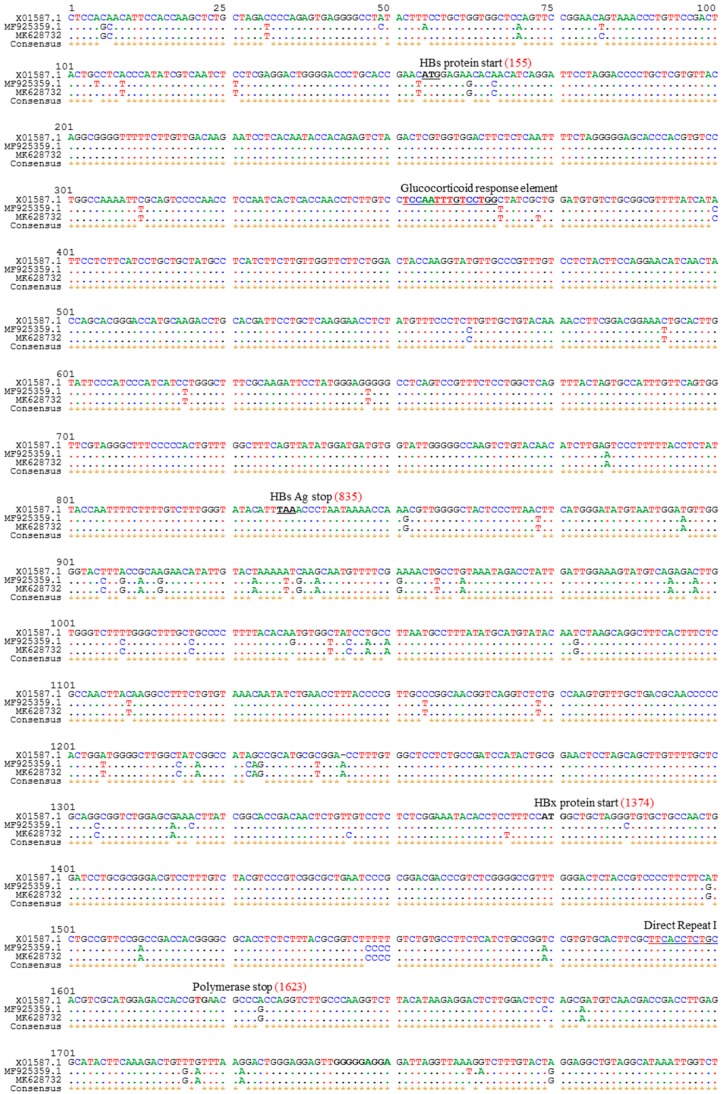
Alignment of HBV full genome sequence with that of reference strains. The HBV complete genome sequence identified in this study was aligned with reference strains MF925359.1 and X01587 and characterized. Mutational analysis was performed by comparing the present sequence with that of the reference HBV genomes.

**Figure 3 ijms-21-00546-f003:**
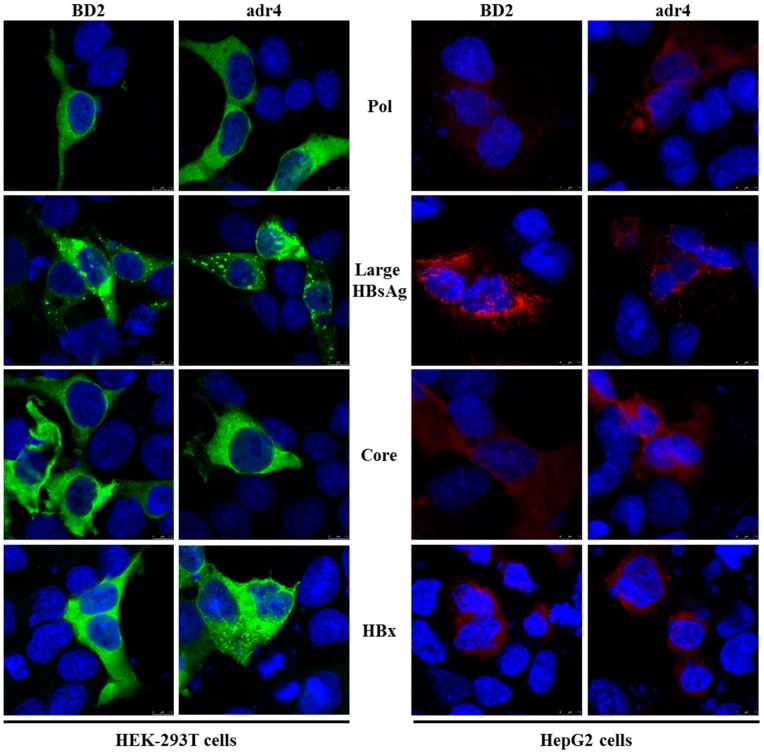
Expression analysis of HBV proteins by immunofluorescence analysis. HEK-293T and HepG2 cells were transfected with pCMV-Myc carrying specific HBV genes. Cells were fixed, permeabilized, and stained with Myc tag Ab at 48 h after transfection, followed by an Alexa Fluor 488-conjugated anti-mouse IgG (Green) for HEK-293T cells and Alexa Fluor 546-conjugated anti-mouse IgG (Red) for HepG2 cells. The cell nuclei were stained with DAPI (blue). The experiment was performed at least three independent times, and one representative data set was presented.

**Figure 4 ijms-21-00546-f004:**
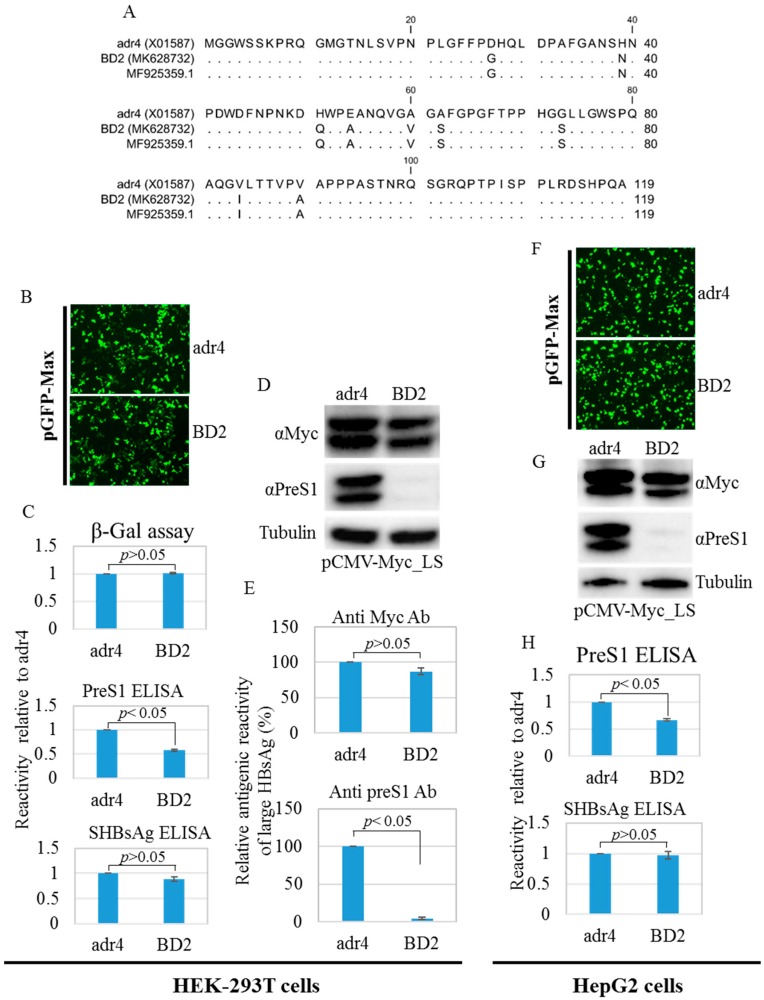
PreS1 region amino acid mutations, western blot analysis, and ELISA. (**A**) Amino acid mutations in the preS1 region compared to the reference HBV strain adr4. Mentioned cells were co-transfected with pCMV-LS carrying a large HBsAg coding sequence and pGFP-Max or pSV-β-Gal and incubated for 48 h. (**B**,**F**) Transfection efficiency was tested by detecting GFP signals from the co-transfected cells. Pictures were taken using EVOS FL Auto with 10×/0.30 magnification. (**C**,**H**) Cells were lysed and subjected to ELISA. ELISA results were normalized to the β-Gal assay. (**D**,**G**) Lysed cells were further subjected to SDS-PAGE followed by western blotting analysis using preS1 and Myc tag antibodies. The β-tubulin used as a loading control. (**E**) Relative antigenic reactivity of large HBsAg by western blot. Data were generated from the densitometry of western blot band intensities detected by preS1, Myc tag, and β-tubulin antibodies. Relative reactivity of large HBsAg was calculated. Each experiment was performed at least three independent times, and results are shown as mean ± SD. *p* ≤ 0.05 was considered as statistically significant.

**Figure 5 ijms-21-00546-f005:**
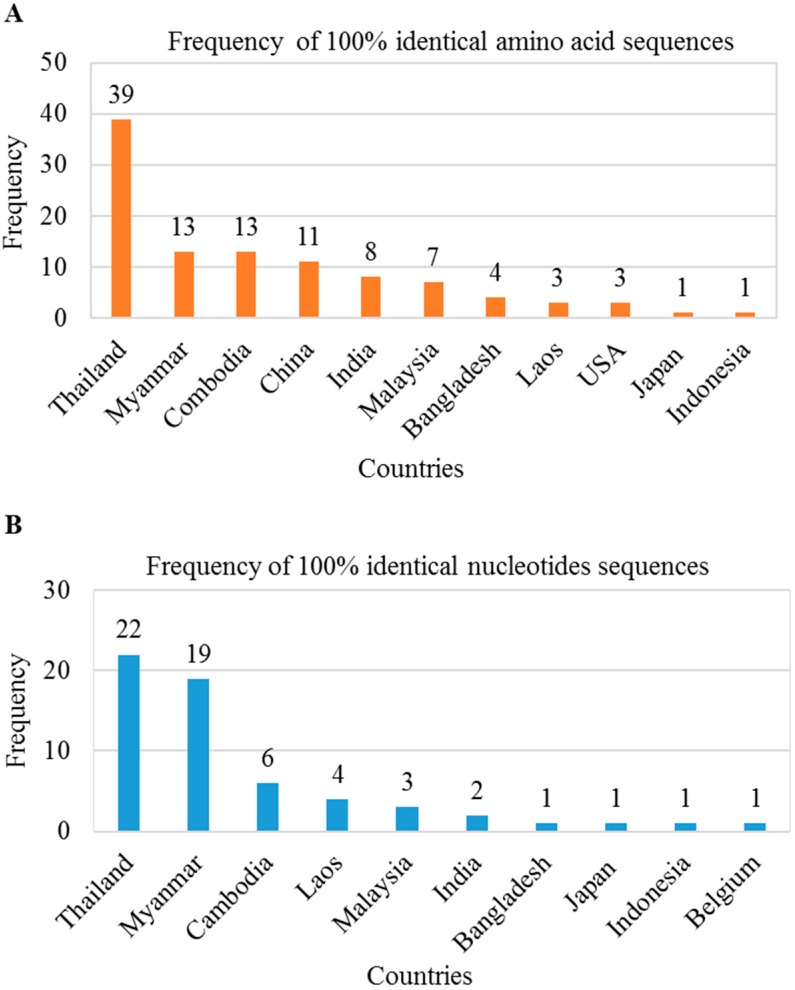
Distribution of silent large HBsAg mutated HBV in Asiatic countries. BLAST searches were performed using 119 amino acids/357 nt of the preS1 region of BD2 genome. A total of 103 amino acid sequences (**A**) and 60 nucleotide sequences (**B**) showed 100% sequence identity.

**Table 1 ijms-21-00546-t001:** List of primers used in this study.

Genes or Purpose	Primer Names	Sequences (5′–3′)
**P**	**EcoRI-Pol-1**	GTGGAATTCGGATGCCCCTATCTTATCAACAC
	**Pol-stop-SalI**	CACGTCGACTCACGGTGGTCTCCATGCGAC
**S**	**EcoRI-HBs preS1**	GTGGAATTCGGATGGGAGGTTGGTCTTCCAAAC
	**HBs-stop-SalI**	CACGTCGACTTAAATGTATACCCAAAGAC
**C**	**EcoRI-Core-1**	GTGGAATTCGGATGGACATTGACCCGTATAAAG
	**Core-stop-SalI**	CACGTCGACTAACATTGAGATTCCCGAG
**X**	**EcoRI-HBx-1**	GTGGAATTCGGATGGCTGCTAGGGTGTGCTG
	**HBx-stop-SalI**	CACGTCGACTTAGGCAGAGGTGAAAAAGTTG
**pCMV_Myc plasmid**	**CMV Pro**	ACGGTGGGAGGTCTATATAAGC
**Real-time quantitative PCR**	**HBs F2**	CTTCATCCTGCTGCTATGCCT
	**HBsR2**	AAAGCCCAGGATGATGGGAT

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
