# Peer review of "PreS1 Mutations Alter the Large HBsAg Antigenicity of a Hepatitis B Virus Strain Isolated in Bangladesh"

_ijms, 2020, doi:10.3390/ijms21020546_

Round 1

Reviewer 1 Report

This manuscript describes the complete sequence analysis of a new HBV isolate from an acutely infected Bangladeshi patient. The authors show that mutations in the pre-S1 region of the large HBsAg alter its detectability by commercial antibodies used in diagnostic kits. These findings are not novel enough to warrant publication in their current form.

Results shown in Fig 1 report routine methods that can be described in the text, but do not require inclusion of a figure.

Fig 4: The report of the subcellular distribution of viral proteins in HEK-293T cells transfected with expression plasmids for individual viral proteins is of limited value. First, the proteins are highly overexpressed and it is known that expression levels can affect the subcellular localization of these proteins. Second, the viral proteins are expressed individually, and not in the presence of all viral proteins, where viral particle assembly could affect localization.

It would be more meaningful to generate an expression plasmid containing a greater than full length HBV genomic copy (commonly used in this sort of study) where these proteins could be expressed together with all viral proteins. Certainly, the individual proteins could no longer be detected with a Myc-antibody, but viral protein specific antibodies are available. The studies should be done in a hepatocyte cell line, not HEK-293T cells, and effects of expression level should be monitored.

Fig 5: The authors report using both ELISA and western blotting to determine the reactivity of large HBsAg from their newly isolated HBV (BD2). Again, they use lysates from HEK-293T cells transfected with a large HBsAg expression plasmids and again, this is a poor choice for these analyses. Overexpression of this single protein, in the absence of other viral proteins, could significantly affect its conformation/aggregation and results may not accurately reflect detection by ELISA from more clinically relevant samples. The western blot (presumably from SDS-PAGE, although this is not described) showed little to no reactivity of the BD2 large HBsAg. This is not unexpected when there are numerous changes in the preS1 region that are likely to affect the antigenic epitope.

Minor edits:

p. 2, line 89: clones for each ORF were (not was) selected

p.14, line 335: "evolutionally" should be "evolutionarily"

This manuscript highlights the important issue of the need for testing antibodies for their reactivity against viral proteins isolated from different strains of virus, especially in clinical, diagnostic, and epidemiological settings. However, results reported in these studies add little new knowledge in that area.

Author Response

Response to Reviewer 1 Comments

Thank you very much for reviewing our manuscript. We really appreciate your suggestions and comments which significantly increased the quality of our manuscript.

Point 1: This manuscript describes the complete sequence analysis of a new HBV isolate from an acutely infected Bangladeshi patient. The authors show that mutations in the pre-S1 region of the large HBsAg alter its detectability by commercial antibodies used in diagnostic kits. These findings are not novel enough to warrant publication in their current form.

Response 1: Actually, to our knowledge, this is the first report investigating changes in large HBsAg antigenicity due to preS1 mutations from clinical case and this mutated preS1 sequence has been identified in several Asian countries which is the major concern. Therefore, it is important to report these mutations for designing new/modify diagnostic kits and will be helpful for preS1 based vaccine development. We also believe that this study will be helpful for further research regarding pathobiology of HBV with mutated preS1 region (As mutations also found in NTCP binding site in this study).

Point 2: Results shown in Fig 1 report routine methods that can be described in the text, but do not require inclusion of a figure.

Response 2: We excluded the figure 1 from the manuscript according to your suggestion.

Point 3: Fig 4: The report of the subcellular distribution of viral proteins in HEK-293T cells transfected with expression plasmids for individual viral proteins is of limited value. First, the proteins are highly overexpressed and it is known that expression levels can affect the subcellular localization of these proteins. Second, the viral proteins are expressed individually, and not in the presence of all viral proteins, where viral particle assembly could affect localization.

Response 3: According to your suggestions, we analyzed the cytoplasmic distribution of viral proteins in HepG2 cells (Data added in figure 3 and described in the results section). Here, we just showed the expression and cytoplasmic distribution with the reference strain adr4. However, previous report also showed HBV proteins (such as Pol or HBsAg) can be expressed in non-hepatic cells and subcellular distribution has not been affected. The viral protein expression distribution also independent of viral replication (Unchwaniwala et al 2016; J Virol.12;90(19):8705-19. doi: 10.1128/JVI.01229-16).

Point 4: It would be more meaningful to generate an expression plasmid containing a greater than full length HBV genomic copy (commonly used in this sort of study) where these proteins could be expressed together with all viral proteins. Certainly, the individual proteins could no longer be detected with a Myc-antibody, but viral protein specific antibodies are available. The studies should be done in a hepatocyte cell line, not HEK-293T cells, and effects of expression level should be monitored.

Response 4: Thank you very much for such an important suggestion. We are planning to design the next project where, we will prepare a replication competent plasmid using this preS1 mutated HBV genome. We will prepare the mutant virus in hepatocyte based cell lines (HepG2 or Huh7 cell lines) by transfection and then will analyze the infectivity of this mutant virus in HepG2-NTCP cells. We will be able to find out the subcellular (specific organelles such as mitochondria, ER etc) localization of viral proteins in infection condition. We also have plan to check the effect of these preS1 mutations with NTCP (cellular receptor for HBV entry) binding. Hopefully, we will be able to explore in detail in our future manuscript.

Point 5: Fig 5: The authors report using both ELISA and western blotting to determine the reactivity of large HBsAg from their newly isolated HBV (BD2). Again, they use lysates from HEK-293T cells transfected with a large HBsAg expression plasmids and again, this is a poor choice for these analyses. Overexpression of this single protein, in the absence of other viral proteins, could significantly affect its conformation/aggregation and results may not accurately reflect detection by ELISA from more clinically relevant samples.

Response 5: According to your suggestions, we analyzed the alteration of large HBsAg antigenicity in HepG2 cells (Results added in Figure 4F, G, and H). May be in some cases it may happen as you said. But single viral protein of HBV could be over expressed in non-hepatic cells for western blot and ELISA based analysis showed in our previous report (Hossain and Ueda 2017; Plos One; https://doi.org/10.1371/journal.pone.0167871. These kind of analyses also performed by several groups ([Alexander et al., 2014; Journal of Hepatology, 61(4): 867-875; https://doi.org/10.1016/j.jhep.2014.05.018], [Melegari et al., 1998; Hepatology.  27(2):628-33; https://doi.org/10.1002/hep.510270243], [Mirian et al 2016, Res Pharm Sci. 2016 Oct; 11(5): 366–373, 10.4103/1735-5362.192485], [Pan et al 2017; Pharmazie.  72(9):550-554. doi: 10.1691/ph.2017.7594]).

Point 6: The western blot (presumably from SDS-PAGE, although this is not described) showed little to no reactivity of the BD2 large HBsAg. This is not unexpected when there are numerous changes in the preS1 region that are likely to affect the antigenic epitope.

Response 6: These mutations could affect the antigenic epitope. We reported it for first time from clinical case in this manuscript. These kind of preS1 mutations affecting antigenicity are circulating in many Asian countries which is the major concern. It is important to report these mutations for designing new/modify diagnostic kits and will be helpful for preS1 based vaccine development.

Point 7: p. 2, line 89: clones for each ORF were (not was) selected

Response 7: Corrected in the manuscript (Lines: 91, 268).

Point 8: p.14, line 335: "evolutionally" should be "evolutionarily"

Response 8: Corrected in the manuscript (Lines: 18 and 344)

Reviewer 2 Report

The manuscript presents an analysis of HBV proteins structure from chronic hepatitis B patients from Bangladesh, and reports that several amino acid substitutions in PreS1 domain of the surface antigen diminishes recognition by the respective antibodies.

The text is quite clearly written, and almost all conclusions are backed up with experimental data. However, there are several questions to the authors and the data.

The cloning of HBV genes was performed with TaKaRa ExTaq polymerase. Yet this is a a bit more precise polymerase compared to the parental Taq (four fold decrease in incorrect nucleotide incorporation), the rate of nucleotide substitutions should be still quite high. This puts into question the amino cid substitutions discovered by the authors. and I did not understand if the authors performed sequencing only of the genes cloned into a plasmid? If so, the mutations can reflect the minor heterogenity due to errors of the polymerase. So, the authors are advised to present sequences of the PCR products obtained during amplification of the genes from patients' serum. If the sequences were already from such PCR products, this should be stated in experimental section much more clearly. Localization of HBV proteins was accessed in HEK293T cells. But the virus does not replicate in kidney cells. So why were they chosen? It would be much better to study the localization in any liver-derived cell lines (such as HepG2 or HepaRG, i.e. in lines that are used to develop infectious systems for HBV). And of course the study would greatly benefit from additional staining of organelles to draw conclusions about precise localization of HBV proteins. Now the only conclusion can be that the proteins remain outside nucleus. But there is a question if HBx localizes in mitochondria, or HBs - in the ER. In my opinion, such experiment is easy to perform, and its results will also contribute to the field.

Author Response

Response to Reviewer 2 Comments

Thank you very much for your constructive criticisms and suggestions. It really helped us to improve the quality of this manuscript.

Point 1: The cloning of HBV genes was performed with TaKaRa ExTaq polymerase. Yet this is a a bit more precise polymerase compared to the parental Taq (four fold decrease in incorrect nucleotide incorporation), the rate of nucleotide substitutions should be still quite high. This puts into question the amino cid substitutions discovered by the authors. and I did not understand if the authors performed sequencing only of the genes cloned into a plasmid? If so, the mutations can reflect the minor heterogenity due to errors of the polymerase. So, the authors are advised to present sequences of the PCR products obtained during amplification of the genes from patients' serum. If the sequences were already from such PCR products, this should be stated in experimental section much more clearly.

Response 1: According to your suggestion, we mentioned in the manuscript (Lines: 93-94; 99-101; 281-283). Actually, we also sequenced the full ORFs (P, S, C and X) from the PCR products. Then during cloning to expression plasmid, we selected two (2) correct fragment containing plasmids for each ORF by restriction enzyme digestion. And these two correct cloned plasmids have been further sequenced to check the sequence diversity with sequence from PCR products. However, cloned plasmid showed 100% matched with sequence from direct PCR product which are presented in the data.

Point 2: Localization of HBV proteins was accessed in HEK293T cells. But the virus does not replicate in kidney cells. So why were they chosen? It would be much better to study the localization in any liver-derived cell lines (such as HepG2 or HepaRG, i.e. in lines that are used to develop infectious systems for HBV). And of course the study would greatly benefit from additional staining of organelles to draw conclusions about precise localization of HBV proteins. Now the only conclusion can be that the proteins remain outside nucleus. But there is a question if HBx localizes in mitochondria, or HBs - in the ER. In my opinion, such experiment is easy to perform, and its results will also contribute to the field.

Response 2: We used 293T cells for high transfection ability. However, we analyzed the expressions, cytoplasmic distribution of viral protein and alteration of large HBsAg antigenicity in HepG2 cells according to your suggestion (Results added in figure 3 and Figure 4F, G, and H and described in the results section). Here, we just showed the expression and cytoplasmic distribution with the reference strain adr4. Actually, we are planning to design the next project where, we will prepare a replication competent plasmid using this preS1 mutated HBV genome. We will prepare the mutant virus in hepatocyte based cell lines (HepG2 or Huh7 cell lines) by transfection and then will analyze the infectivity of this mutant virus in HepG2-NTCP cells. We will be able to find out the subcellular (specific organelles such as mitochondria, ER etc) localization of viral proteins in infection condition as you know there is a controversy of localization of HBx into mitochondria (https://jvi.asm.org/content/82/14/6798) and nucleus (https://www.ncbi.nlm.nih.gov/pubmed/19464721; https://jvi.asm.org/content/jvi/93/16/e00248-19.full.pdf); and HBsAg of course into the ER but some fractions may go into mitochondria (https://www.ncbi.nlm.nih.gov/pubmed/23275097). We also have plan to check the effect of these preS1 mutations with NTCP (cellular receptors for HBV entry) binding. Hopefully, we will be able to explore in detail in our future manuscript.

Round 2

Reviewer 1 Report

The authors and this reviewer disagree about the value of detection and localization studies of massively over-expressed viral proteins, sometimes in irrelevant cell types. There are elements of this manuscript that merit reporting, particularly the confirmation of a clinical isolate that is not detected by antibodies used in commercial testing kits. However, that result is not surprising and undoubtedly reflects sequence changes in the epitope. It is not clear what broader conclusion the authors are drawing from individually cloned and over-expressed viral proteins from this stain in transfected cells. 

Reviewer 2 Report

The authors answered two my major concerns (sequencing of PCR products and expression of HBV protein in liver cells). So, the manuscript has been improved.